# The Safety and Cost Analysis of Outpatient Laparoendoscopy in the Treatment of Cholecystocholedocholithiasis: A Retrospective Study

**DOI:** 10.3390/jcm13020460

**Published:** 2024-01-14

**Authors:** José Sebastião dos Santos, Rafael Kemp, Alicia Guadalupe Mendoza Orquera, Alberto Facury Gaspar, Jorge Resende Lopes Júnior, Lucas Tobias Almeida Queiroz, Víctor Antônio Peres Alves Ferreira Avezum, José Celso Ardengh, Ajith Kumar Sankarankutty, Leonardo Santos Lima

**Affiliations:** 1Department of Surgery and Anatomy, Faculty of Medicine of Ribeirão Preto, University of São Paulo (FMRP-USP), São Paulo 05508-220, Brazil; rkemp@fmrp.usp.br (R.K.); ajith@fmrp.usp.br (A.K.S.); 2Postgraduate Program, Department of Surgery and Anatomy, Faculty of Medicine of Ribeirão Preto, University of São Paulo (FMRP-USP), São Paulo 05508-220, Brazil; alikatherin@gmail.com (A.G.M.O.); afgaspar@hcrp.usp.br (A.F.G.); ltaqueiroz@hcrp.usp.br (L.T.A.Q.); victoravezum@hcrp.usp.br (V.A.P.A.F.A.); leonardolima@hcrp.usp.br (L.S.L.); 3University Hospital, Faculty of Medicine of Ribeirão Preto, University of São Paulo (FMRP-USP), São Paulo 05508-220, Brazil; jresendejr@hcrp.usp.br (J.R.L.J.); jcelso@uol.com.br (J.C.A.); 4Digestive Endoscopy Service, Hospital Moriah, São Paulo 04084-002, Brazil; 5Department of Diagnostic Imaging, Federal University of São Paulo, São Paulo 04021-001, Brazil

**Keywords:** cholecystocholedocholithiasis, laparoscopic, choledocholithiasis, endoscopic sphincterotomy, intraoperative endoscopy, laparoendoscopy, outpatient surgery, cost analysis

## Abstract

Introduction: The advantages of single-stage treatment of cholecystocholedocholithiasis are well established, but the conditions for carrying out treatment on an outpatient basis require a review of concepts and practices of medical corporations. Objective: To evaluate the practice of treating cholecystocholedocholithiasis by laparoendoscopy on an outpatient basis with cost analysis. Method: A retrospective study was conducted on patients with cholecystocholedocholithiasis treated by combined laparoscopic cholecystectomy and endoscopic choledocholithotomy from January 2015 to January 2019. After collecting data from physical and digital medical records, the patients were divided into two groups—AR (*n* = 42)—ambulatory regimen and HR (*n* = 28)—hospitalization regimen—which were compared in terms of demographic, clinical and treatment variables and their results, as well as in terms of costs. Results: The mean age of the AR group was lower than that of the HR group and the physical status of the AR patients was better when assessed according to the American Society of Anesthesiologists (ASA) (*p* = 0.01). There was no difference between groups regarding the risk of choledocholithiasis (*p* = 0.99). For the AR group, the length of stay was shorter: 11.29 h × 65.21 h (*p* = 0.02), as was the incidence of postoperative complications assessed by applying the Clavien–Dindo classification: 3 (7.1%) × 11 (39.2%) (*p* < 0.01). The total mean costs were higher for the HR group (USD 2489.93) than the AR group (USD 1650.98) (*p* = 0.02). Conclusion. Outpatient treatment of cholecystocholedocholithiasis by laparoendoscopy is safe and viable for most cases, has a lower cost and can support the reorientation of training and practice of hepatobiliary surgeons.

## 1. Introduction

The treatment of cholecystocholedocholithiasis in a single surgical time in patients by the approach to the main bile duct by videolaparoscopy alone or in combination with endoscopic retrograde cholangiopancreatography (ERCP) represents an evolutionary milestone in minimally invasive treatment, with good clinical results [1,2,3] and lower costs [4,5,6]. However, the combination of accesses, techniques and tactics and the assistential processes are still a subject of debate among surgeons and endoscopists.

Treatment in a single time by a laparoscopic approach for the treatment of bile stones with an in situ gallbladder has the same advantages as standard surgery but reduces the number of procedures per patient and the duration of hospitalization, eliminates the underlying risk of endoscopy sphincterotomy and leaves the Oddi sphincter intact [7].

On the other hand, endoscopic treatment of main bile duct stones is considered to be the best option in the era of minimally invasive surgery and its performance in the pre- and postsurgical phases (videolaparoscopic cholecystectomy) has been well established. However, routine access to the bile duct by ERCP involves the execution of an unnecessary number of procedures and complications, whereas the combination of videolaparoscopic cholecystectomy and endoscopic sphincterotomy has shown better cost-effectiveness results when compared to other therapeutic possibilities [8,9].

Intraoperative cholangiography during laparoscopy for the identification of bile duct stones has the advantage of reducing the need for more accurate preoperative exams such as magnetic resonance cholangiography, echoendoscopy and ERCP, also permitting the removal of stones by laparoscopy or endoscopy, with shorter hospitalization and reduced hospital costs. However, the identification of patients who might benefit from the combination of videolaparoscopic cholecystectomy and endoscopic sphincterotomy for the treatment of cholelithiasis plus coledocholithiasis, on an outpatient basis, plus the cost analysis, is an original approach that motivated the disclosure of the present study.

Therefore, the present study aims to evaluate the profile of patients undergoing laparoendoscopic treatment (LE), comparing treatment regimens (outpatient vs. inpatient) with cost analysis. It is expected that patients with controlled comorbidities have a lower rate of complications and lower costs in the outpatient setting.

## 2. Materials and Methods

The present study was conducted in a tertiary public teaching hospital located in the interior of São Paulo, Brazil, associated with the Ribeirão Preto Medical School of the University of São Paulo, considered as a national reference center in the surgical and endoscopic treatment of biliopancreatic diseases. The study was registered on the Plataforma Brasil (CAAE 88864818.2.1001.5440) and submitted to the Institutional Ethics Committee of the University Hospital, Faculty of Medicine of Ribeirão Preto, University of São Paulo (opinion 2.849.107). A retrospective study was carried out including all patients with cholecystocholedocholithiasis who underwent single-stage treatment through laparoendoscopy (LE), with the combination of laparoscopic cholecystectomy, intraoperative cholangiography and endoscopic removal of the biliary stone, operated from January 2015 to January 2019.

After collecting data from physical and digital medical records, the patients were divided into two groups—AR (*n* = 42), ambulatory regimen, and HR (*n* = 28), hospitalization regimen—which were compared in terms of demographic, clinical and treatment variables and their results, as well as in terms of costs.

The technique in the study consisted of a combination of LC and endoscopic sphincterotomy (ES) for the simultaneous treatment of cholelithiasis and extraction of main bile duct (MBD) stones in a single anesthetic–surgical procedure [10]. Antibiotic prophylaxis with cefazolin was performed in each case and each patient was kept in dorsal decubitus throughout the procedure (American position). LC was performed according to the traditional technique and the patient was set up in the American position model and trancystic intraoperative cholangiography (IOC) was performed in all cases [11]. The IOC technique was carried out using 300 mg/IU iodinated contrast at 50% dilution and at body temperature. The endoscopic procedure was performed using an Olympus^®^ TJF-C180 duodenoscope with a 0.025Fr (French) hydrophilic catheter and a cholangiography catheter (sphincterotome), both from Boston Scientific*^®^* (Marlborough, MA, USA). Access to the MBD was obtained by two pathways, the retrograde transpapillary one (by conventional ERCP), and the anterograde one according to the rendezvous laparoendoscopy technique (LERV) [12]. After access to the bile duct, an endoscopic sphincterotomy (ES) was performed, followed by stone extraction and bile duct clearance. The effectiveness of treatment was checked in terms of full clearance of the bile duct, defined by successful stone removal with no residues and confirmed by the injection of contrast at the end of the procedure [10].

Data analysis was based on clinical variables regarding the physical health status of the patient, the risk of choledocholithiasis and the determination of effectiveness, and on effect variables for the assessment of hospital costs such as surgical time, type of hospital stay, and expenses related to hospital stay, medications, exams, and procedures. The Clavien–Dindo classification of surgical complications was used for the standardization of the results according to the information in the medical records [13]. The in-hospital stay or permanence regimen was defined on the basis of the patient’s postoperative destination, with patients discharged within 24 h after admission to the hospital being considered as treated on an AR basis (with or without an overnight hospital stay), and subjects discharged after more than 24 h being considered to be treated on an HR basis. The AR and HR groups were analyzed comparatively according to the main objective of the present study.

The cost analysis was based on absorption cost, which considers direct costs (medication and complementary tests) and indirect costs (cost per hour in the operating room and recovery room after anesthesia, as well as the daily bed cost intensive care unit) obtained according to the criteria of apportionment of expenses with materials, costing and maintenance of environments and professional fees established for the reality of the institution itself [14]. The micro-costing of the diagnosis and treatment for each patient was elaborated by listing the medications and complementary exams contained in the medical records and calculating the expenses of the institution itself with the acquisition of the items, added to the length of stay in the various sectors of the hospital. Average costs in US dollars for 2018 were considered for each of these variables. The average value in Brazilian reais (BRL) was 3.88 for each dollar (USD). For the cost analysis study, the differences between the average costs of the entire treatment of the AR and HR groups were compared.

The data were first described as absolute and percent frequencies (qualitative variables) and as means and standard deviation (quantitative variables). All comparisons involving quantitative variables were analyzed by the Student *t*-test, while the comparisons involving qualitative variables were analyzed by the Fisher exact test, with the level of significance set at *p* < 0.05. The statistical software SPSS version 21 was used for data analysis and for graph elaboration.

## 3. Results

Regarding the eligibility criteria, all cases with cholelithiasis and risk of choledocholithiasis were selected, leaving 99 eligible cases. Of these were also 28 excluded cases of videolaparoscopic cholecystectomy (LC) with normal intraoperative cholangiography, and 1 case requiring conversion to laparotomy. Finally, 70 cases were selected in the present study (Figure 1).

Of the 70 patients included in the study, 67.1% (*n* = 47) were females and 32.9% (*n* = 23) were males; the mean age was 47.5 years. According to the American Society for Gastrointestinal Endoscopy (ASGE) criteria [3], there were no cases at low risk of choledocholithiasis and according to the American Society of Anesthesiologists (ASA) health status classification, no patient was ASA IV. In total, 60% of the patients (*n* = 42) were treated as AR patients and 40% (*n* = 28) with more than 24 h of hospitalization as HR patients. Considering only AR patients, 69% (*n* = 29) were treated and discharged after an average observation period of 7.5 h, while still in the anesthetic recovery room, i.e., without hospitalization at night, while only 31% (*n* = 13) required an overnight stay and were discharged less than 24 h after admission (Table 1).

There was no difference in gender distribution between patients in the AR and HR groups (*p* = 0.61), while the mean age of AR patients was lower (43.52 × 53.46—*p* = 0.03). Regarding physical health status (ASA), there was a predominance of ASA I and II in the AR group (*n* = 38; 90.5%), and a predominance of ASA II and III patients in the HR group (*n* = 24; 85.7%) (*p* < 0.01), as shown in Table 1.

Surgical time including LC and ERCP (minutes) from the beginning to the end of surgery was similar for the two groups (*p* = 0.16), as were the techniques of access to and stone extraction from the bile duct. The technique of access to the MBD was mainly based on the conventional approach (sphincterotomy) in 60% of cases (*n* = 42), followed by rendezvous access in 17.3% (*n* = 17). The effectiveness of treatment did not differ between AR and HR, with only one HR case of incomplete clearance of the bile duct (Table 1). According to the Clavien–Dindo scale, the incidence of postoperative complications was higher in the HR group (*p* < 0.01) (Table 1).

The overall cost including medications, time in post-anesthesia recovery, exams and medications was lower for the AR group (*p* = 0.02), with differences observed in hospital permanence and hospitalization (*p* = 0.02) and in the surgical procedures (*p* = 0.01) (Table 2).

The total mean costs were higher for the HR group (USD 2489.93) than the AR group (USD 1650.98) (*p* = 0.02). The absence of complications positively affected the total cost of hospitalization, with estimated savings of USD 681,98 (1182.15; 181.63), with 95% CI and *p* = 0.01 (Figure 2). The cost with ERCP in both of the groups was USD 303,1 and this value was computed with the cost total. (Table 2).

## 4. Discussion

Choledocholithiasis is a worldwide clinical problem with a prevalence of 10 to 20% among patients with symptoms due to cholelithiasis [15]. Choledocholithiasis may involve complications such as cholangitis and pancreatitis which require increased hospitalizations, more elaborate imaging exams and therapeutic procedures, as well as increased morbidity–mortality and costs [16].

The development of new endoscopic technologies and the expansion of videolaparoscopic surgery have permitted the combined use of these routes of access in a single surgical time for the treatment of choledocholithiasis with the gallbladder in situ [17,18,19]. Among the treatment modalities previously compared with good results [1,2,3], hybrid treatment (laparoendoscopic) and fully laparoscopic treatment with exploration of the bile ducts have shown the lowest rate of complications [1,2,3] and costs^,^ [17,18,19,20]_._

The main determinant influencing the cost of choledocholithiasis treatment is the length of stay. Within this context, the identification of patients who may benefit from the combined use of videolaparoscopic cholecystectomy and endoscopic sphincterotomy for the treatment of cholelithiasis plus choledocholithiasis in a single surgical time and in an ambulatory regimen followed by the cost analysis is an original approach which motivated the disclosure execution of the present study.

The procedures were performed by a single team with standard hepatobiliary surgical training and advanced laparoscopic and endoscopic training, in contrast to the format used in some other studies [8,9,20,21], but already resembling what is observed in some American centers, where surgically trained endoscopists perform both methodological approaches [18].

IOC was performed successfully in all cases for the intraoperative confirmation of choledocholithiasis, with a 95.5% success rate for the cannulation of the main bile duct [22,23]. This practice can potentially simplify the investigation of choledocholithiasis of habitual presentation, without the need for more complex exams such as cholangiography by magnetic resonance and endoscopy and echoendoscopy.

The laparoendoscopic *rendezvous* technique (LERV), carried out with the anterograde introduction of a transcystic guide wire, facilitates the cannulation of the papilla during the endoscopic procedure and has been increasingly used by multidisciplinary teams [24,25,26,27], possibly reducing the risk of post-ERCP pancreatitis [28,29] as a function of easy cannulation [20]. In the present study, this technique was used in 12.9% of cases (*n* = 9). Among the patients studied (*n* = 70), there were only three cases of pancreatitis associated with the injection of contrast into the MBD, but not with the route of access.

The position of the patient during treatment is an important point of discussion since it may cause difficulty of access to the bile duct. The supine position can increase the degree of difficulty in the positioning of the duodenoscope but is effective and safe for the access to the bile duct with the patient intubated [12,18]. In the present study, none of the patients required a change in position for access to the bile duct.

The treatment of cholelithiasis and choledocholithiasis in a single surgical time with IOC and videolaparoscopic cholecystectomy and endoscopic choledocholithotomy potentially reduces the number of visits to the hospital and ERCP, the need for two anesthetic procedures, the risk of pancreatitis, the time of hospitalization and the cost compared to treatment performed in two stages [28,29]. Additionally, compared to treatment in a single stage by the laparoscopic route, it guarantees better duct clearance than trancystic exploration [19] and does not require drainage of the abdominal cavity and bile duct.

In this analysis, the procedure was effective in 98.5% of cases, with resolution and full clearance of the bile duct. The success rate of intraoperative ERCP oscillates from 69.2% to 100%, with a mean of 92.3% [30] and the efficacy of bile duct clearance with intraoperative ERCP is 96.3%, equivalent to that of preoperative ERCP (96.9%) [18]. Additionally, a systematic review has shown a 2.27-times higher risk of complications after preoperative ERCP compared to intraoperative ERCP [4], supporting the practice adopted here.

The morbidity was 20%, with no report of death among the 70 patients investigated. Morbidity was lower in the AR group (7.14%) compared to the HR group (39.29%), where 21.3% of patients were classified as ASA III. The mean morbidity reported in the literature is 5.1% (range: 0-19%) and mortality is rare (0.37%) [19,31].

The mean time of permanence in the operating room was 128.8 min (SD: 39.31) for the 70 patients studied, with no difference between the AR and HR groups (*p* = 0.16). This finding is similar to that reported in the literature, where the mean duration of the endoscopic procedure is 35 min, with a duration of surgery up to 104 min, for a total of 135 min. There were no cases of conversion to open surgery, exceeding the expectations reported in the literature, where the mean rate of conversion to open surgery is 4.7% [10].

In most studies, treatment by hybrid access in a single surgical time was performed under HR, with no emphasis on AR. Among treated patients, 60% (*n* = 42) were assigned to AR, with 69% of them (*n* = 29) being treated and discharged after a mean time of observation of 7.5 h in the anesthesia recovery room, i.e., without an overnight hospital stay. An extensive literature review has shown that the mean time of hospitalization using intraoperative ERCP was 2.8 days per patient, pointing out the reduction of costs [32], which can be maximized with the strategy used in the present study.

The first signs and symptoms of pancreatitis or cholangitis after ERCP are known to occur during the first 4 h and up to 90% of the complications can be detected up to 6 h after the procedure [33]. This information guided the time of hospital stay and the telephone contact provided by the health team on the day after the operation, since 92.86% (*n* = 39) of the AR patients showed no complications. The absence of complications had a positive effect on the reduction of the total cost of hospitalization, with estimated savings of USD 658.34.

The success of ambulatory surgery depends, among other factors, on the training and integration of the assistance team, on appropriate patient selection, and on the systematization of the care process. The combination of laparoscopic cholecystectomy and endoscopic choledocholithotomy for the treatment of choledocholithiasis with cholelithiasis in a single surgical time proved to be safe and feasible in AR for patients younger than 63 years and classified as ASA 1 and 2, with lower costs.

Finally, the total mean costs found in this study in hospitalized patients (USD 2489.93) were considerably lower compared to the costs described in other studies (USD 7988). The method of micro-costing analysis and the difference in salary of health professionals, considering a developing country, may explain this finding [30].

It is important to consider that, despite the methodological rigor, and data collection by two trained investigators with technical knowledge, it is still desirable that future studies can include a prospective design to advance evaluating the present research question. Some limitations were considered in this study, such as the following: the absence of variables related to biliary tract conditions, and the lack of descriptive analysis of gallstones and inflammatory process in the cystic region, which could make the technique difficult, impacting possible complications. As there was no standardization of these variables, we preferred to exclude them from the analysis.

Further studies, especially of a prospective design, are needed for the definition of protocols of preoperative investigation of patients with a moderate or high risk of choledocholithiasis and for the comparison of hybrid laparoendoscopic treatment to fully laparoscopic treatment in a single surgical time, in addition to the adoption of AR whenever possible. The results of such studies may support or reorient the concept and the practice of patient care adopted in this context. They may also subsidize the reorientation of professional training programs for hepatobiliary surgeons with the integrated use of technological resources and the adoption of care protocols that might refine the clinical management of the approach to choledocholithiasis and further improve the relationship between effectiveness, use of resources and costs.

## Figures and Tables

**Figure 1 jcm-13-00460-f001:**
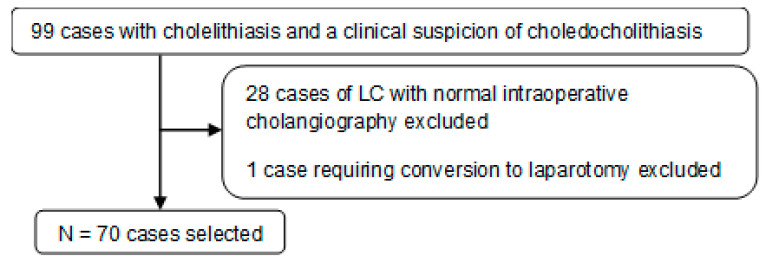
Flow diagram for patient selection.

**Figure 2 jcm-13-00460-f002:**
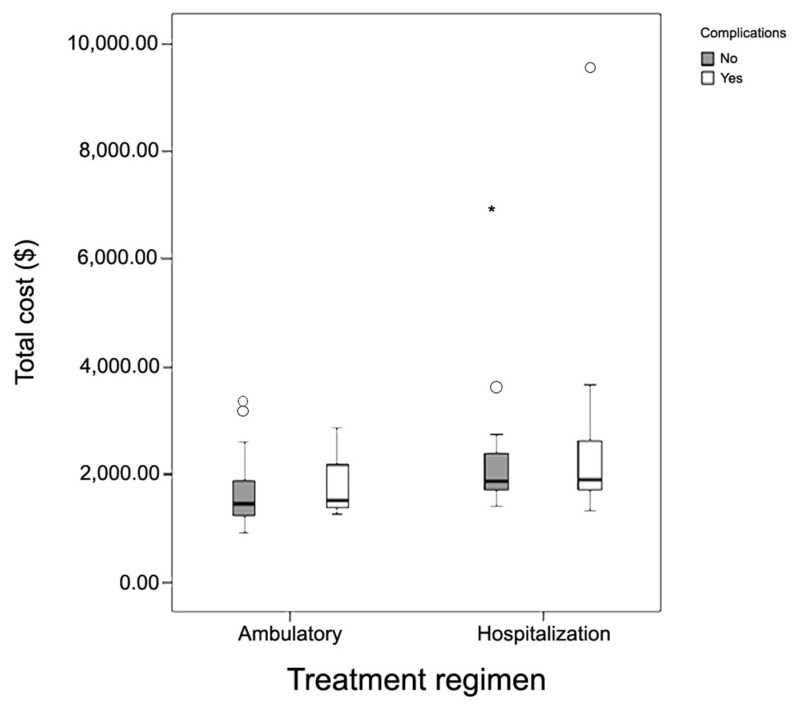
Total cost of the treatment of patients with cholelithiasis and choledocholithiasis performed by laparoendoscopy (LE) on an ambulatory regimen (AR) and hospitalization regimen (HR), with or without complications, from January 2015 to January 2019. Legend: * and circle (outliers).

**Table 1 jcm-13-00460-t001:** Demographic, clinical and therapeutic data of patients with cholelithiasis and choledocholithiasis treated with a laparoendoscopy (LE), on an ambulatory regimen (AR) or a hospitalization regimen from January 2015 to January 2019.

			LE Treatment		
Variables		N = 70	AR = 42	HR = 28	
		Mean (SD)	Mean (SD)	Mean (SD)	*p*
Age (years)		47.5 (19.12)	43.52 (19.04)	53.46 (17.96)	0.03 *
		*n*/%	*n*/%	*n*/%	*p*
Gender	Male	23 (32.9%)	15 (35.7%)	8 (28.6%)	0.61
	Female	47 (67.1%)	27 (64.3%)	20 (71.4%)	
ASA	I	29 (41.4%)	25 (59.5%)	4 (14.3%)	<0.01 *
	II	31 (44.3%)	13 (31%)	18 (64.3%)	
	III	10 (14.3%)	4 (9.5%)	6 (21.4%)	
Risk of choledocholitiasis	High	55 (78.6%)	33 (78.6%)	22 (78.6%)	0.99
	Intermediate	15 (21.4%)	9 (21.4%)	6 (21.4%)	
	Low	0 (0%)	0 (0%)	0 (0%)	
MBD diameter	mm	11.44 (2.1)	11.38 (2)	11.54 (2.27)	0.76
Stone diameter	mm	7.92 (3.25)	7.97 (3.13)	7.84 (3.5)	0.87
Access to the MBD	Sphincterotomy	17 (24.3%)	11 (26.2%)	6 (21.4%)	0.91
	*Rendezvous*	9 (12.9%)	6 (14.3%)	3 (10.7%)	
	Infundibulotomy	42 (60%)	24 (57.1%)	18 (64.3%)	
	Fistulotomy	2 (2.9%)	1 (2.4%)	1 (3.6%)	
Manipulation of the MPD	Yes	16 (22.9%)	9 (21.4%)	7 (25%)	0.78
	No	54 (77.1%)	33 (78.6%)	21 (75%)	
Pancreatography	Yes	2 (2.9%)	1 (2.4%)	1 (3.6%)	0.99
	No	68 (97.1%)	41 (97.6%)	27 (96.4%)	
Surgical time	Minutes	128.76 (39.31)	122.9 (31.02)	137.54 (48.52)	0.16
Lenght of stay	Hours	32.86 (52.38)	11.29 (6.15)	65.21 (71.72)	<0.01
Effectiveness	Yes	64 (98.5%)	41 (100%)	23 (95.8%)	0.37
	No	1 (1.5%)	0 (0%)	1 (4.2%)	
Postoperative complications	None	56 (80%)	39 (92.86%)	17 (60.71%)	<0.01 *
	Wound infection	2 (2.86%)	0 (0%)	2 (7.14%)	
	ERCP pancreatitis	3 (4.29%)	2 (4.76%)	1 (3.57%)	
	Papillary hemorrhage	2 (2.86%)	1 (2.38%)	1 (3.57%)	
	Duodenal perforation	1 (1.43%)	0 (0%)	1 (3.57%)	
	Umbilical hernia	1 (1.43%)	0 (0%)	1 (3.57%)	
	Wound dehiscence	1 (1.43%)	0 (0%)	1 (3.57%)	
	CRI becoming acute	1 (1.43%)	0 (0%)	1 (3.57%)	
	Renal dysfunction	1 (1.43%)	0 (0%)	1 (3.57%)	
	Abdominal collection	2 (2.86%)	0 (0%)	2 (7.14%)	
Clavien–Dindo	Grade I	7 (10%)	1 (2.4%)	6 (21.4%)	< 0.01 *
	Grade II	2 (2.9%)	1 (2.4%)	1 (3.6%)	
	Grade IIIa	1 (1.4%)	0 (0%)	1 (3.6%)	
	Grade IIIb	2 (2.9%)	0 (0%)	2 (7.1%)	
	Grade IVa	0 (0%)	0 (0%)	0 (0%)	
	Grade IVb	1 (1.4%)	0 (0%)	1 (3.6%)	
	Grade V	0 (0%)	0 (0%)	0 (0%)	

* *p* < 0.05. AR—ambulatory regimen; HR—hospitalization regimen; ASA—physical health status according to the American Association of Anesthesiology; MBD—main bile duct; MPD—main pancreatic duct.

**Table 2 jcm-13-00460-t002:** Cost analysis of the treatment of patients with cholelithiasis and choledocholithiasis using laparoendoscopy (LE), in an ambulatory regimen (AR) and hospitalization regimen (HR), from January 2015 to January 2019.

Variable	LE Treatment		Estimated Difference (95% CI)	*p*-Value
	AR (*n* = 42)	HR (*n* = 28)		
	Mean (SD)	Mean (SD)		
Cost of hospitalization	305.03 (458.38)	874.12 (1148.38)	−569.08 (−1033.59; −104.58)	0.02 *
Cost of medication	9.88 (16.84)	46.18 (92.59)	−36.3 (−72.53; −0.08)	0.05
Cost of surgery	930.15 (206.55)	1133.27 (445.56)	−203.12 (−360.54; −45.7)	0.01 *
Cost of the ARR	56.91 (47.95)	52.22 (47.41)	4.7 (−18.54; 27.94)	0.69
Cost of exams	45.91 (89.34)	81.05 (146.56)	−35.14 (−91.37; 21.1)	0.22
Total cost	1650.98 (586.91)	2489.93 (1715.1)	−838.95 (−1525.09; −152.8)	0.02 *

* *p* < 0.05. AR—ambulatory regimen; HR—hospitalization regimen; ARR—anesthesia recovery room.

## Data Availability

The data that support the findings of this study are available on request from the corresponding author, [J.S.d.S.]. The data are not publicly available due to containing information that could compromise the privacy of research participants.

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
