# Peer review of "The Safety and Cost Analysis of Outpatient Laparoendoscopy in the Treatment of Cholecystocholedocholithiasis: A Retrospective Study"

_jcm, 2024, doi:10.3390/jcm13020460_

Round 1
Reviewer 1 Report
Comments and Suggestions for Authors
This is a local audit of service and cost analysis study. The set up is a tertiary hospital in Brazil.
The study design is cross-sectional retrospective and this is appropriate for the question asked. The study analyses ambulatory vs hospitalised patients treated with combined laparoscopic cholecystectomy & endoscopic choledocholithotomy over a time period 4 years.
The study design is appropriate for a service evaluation and cost analysis. It would be useful to see the long-term follow up of the cohort.
Would suggest reporting median (IQR) instead of mean age if data is non-parametric.
I do not understand what the authors mean by 'no difference in gender' - is this an equal gender distrubtion or M:F 1:1? Suggest amending this.
Please correct 'american studies' to 'American studies' or 'US-based studies'. Otherwise, written in clear English.
Comments on the Quality of English Language
A few minor edits.
Author Response
Response to review 1:
1) “I would suggest reporting median (IQR) rather than mean age if the data are not parametric”.
We did not report the median because of the presentation of the data as percentages (as shown in the table) and we reported the use of the mean for the variable "age" because it presented a normal distribution.
2) “I do not understand what the authors mean by 'no difference in gender' - is this an equal gender distrubtion or M: F 1:1? Suggest amending this”.
Correct, let's make it clear: “There was no difference in gender distribution between patients in the AR and HR groups (p=0.61).
3)“Please correct 'american studies' to 'American studies' or 'US-based studies'. Otherwise, written in clear English”.
Of course, let's adopt the expression American studies.

Reviewer 2 Report
Comments and Suggestions for Authors
I have read with great interest the manuscript entitled "Outpatient Treatment of Cholecystocholedocholithiasis by Laparoendoscopy and the Motivation for Reviewing Training in Hepatobiliopancreatic Surgery" by Santos et al. Despite clinical significance, there are some apprehensions I have to point out to arouse discussion on the heated and flourishing research topic.
Major comments:
1. Title: "The Safety and Cost-effectiveness analysis of Outpatient Laparoendoscopy in the Treatment of Cholecystocholedocholithiasis: A Retrospective Study" is recommended.
2. Abstract: In the sentence of "A cross-sectional retrospective study ...", the word "cross-sectional" should be deleted, so as the others in the manuscript. The nature of cross-sectional design should be grasped by the authors.
3. Materials and Methods:
1) The ethical approval number and clinical trial registration number of the present study should be provided.
2) "Regarding the eligibility criteria, all cases with cholelithiasis and risk of choledocholithiasis were selected, leaving 101 eligible cases. Of these, were also excluded 28 cases of videolaparoscopic cholecystectomy (LC) with normal intraoperative cholangiography, and 1 case requiring conversion to laparotomy. Finally, 70 cases were selected in the present study (Figure 1)." should be transferred to the section of Results.
3) A more complete and delicate flow diagram should be resubmitted by the authors.
4. Results: The inadequate sample size would compromise the robustness of the conclusion to some extent.
5. The sections of "Author Contributions", "Institutional Review Board Statement", "Informed Consent Statement", "Data Availability Statement", and "Acknowledgments" should be completed.
Comments on the Quality of English LanguageAcceptable.
Author Response
1) “Title: "The Safety and Cost-effectiveness analysis of Outpatient Laparoendoscopy in the Treatment of Cholecystocholedocholithiasis: A Retrospective Study" is recommended”.
The title suggestion can be accepted almost in full: "The Safety and Cost analysis of Outpatient Laparoendoscopy in the Treatment of Cholecystocholedocholithiasis: A Retrospective Study".
Therefore, instead of employing the concept of cost-effectiveness, we recommend only cost analysis.
Cost-effectiveness analysis is best applied to compare treatment modalities, such as laparoscopic versus laparoendoscopic treatment for cholecystocholedocholithiasis and, in the present study, we compared patients undergoing the same treatment modality and their costs according to the possibilities of identifying two groups: one which can be treated on an outpatient basis and another which requires hospitalization.
2) Abstract: In the sentence of "A cross-sectional retrospective study ...", the word "cross-sectional" should be deleted, so as the others in the manuscript. The nature of cross-sectional design should be grasped by the authors.
Perfectly, the adjustment was made.
3) Materials and Methods: The ethical approval number and clinical trial registration number of the present study should be provided.
Perfectly, the adjustment was made.
4) "Regarding the eligibility criteria, all cases with cholelithiasis and risk of choledocholithiasis were selected, leaving 101 eligible cases. Of these, were also excluded 28 cases of videolaparoscopic cholecystectomy (LC) with normal intraoperative cholangiography, and 1 case requiring conversion to laparotomy. Finally, 70 cases were selected in the present study (Figure 1)." should be transferred to the section of Results.
Perfectly, the adjustment was made.
5) A more complete and delicate flow diagram should be resubmitted by the authors.
Perfectly, the adjustment was made.
6) Results: The inadequate sample size would compromise the robustness of the conclusion to some extent.
The conclusion is based on the experience obtained with the consecutive inclusion of all cases with suspected cholecystocholedocholithiasis treated in the period and, therefore, does not constitute a sample. The discussion highlighted the need to bring together technical and operational resources for the laparoendoscopic treatment modality by a single team and at a single time, as well as the limitations of observational studies. Thus, the conclusion was reached, based on the detailed description of the circumstances of this retrospective analysis of a consecutive series of patients with cholecystocholedocholithiasis and allows to instigate a review of the skills of hepatobiliary surgeons to better meet the needs of patients and healthcare systems.
7) The sections of "Author Contributions", "Institutional Review Board Statement", "Informed Consent Statement", "Data Availability Statement", and "Acknowledgments" should be completed.
Perfectly, the adjustment was made.

Round 2
Reviewer 2 Report
Comments and Suggestions for Authors
I support the publication of the present manuscript.